# Quality Management for Radiation Oncology In-House Software Products

**DOI:** 10.3390/bioengineering12040352

**Published:** 2025-03-29

**Authors:** Sua Yoo, Phillip Antoine, Chunhao Wang, Yang Sheng, Q. Jackie Wu, Joseph Kowalski, Qiuwen Wu, Fang-Fang Yin, William Giles

**Affiliations:** Department of Radiation Oncology, Duke University, Durham, NC 27701, USA

**Keywords:** quality management, in-house software, radiotherapy

## Abstract

(1) Objective: To develop and implement an in-house software quality management (ISQM) program to ensure the continuous, consistent quality of in-house software products. (2) Methods: The ISQM program consists of two key components: quality assurance (QA) and comprehensive documentation. The QA component involves code review, acceptance testing, commissioning, and routine quality checks. Documentation requirements encompass a product report, user manual, QA procedures and reports, release notes, version control, and logs. For each software product, the QA process must be completed, and all required documentation must be finalized before clinical deployment. (3) Results: The ISQM program was successfully developed and retrospectively implemented for existing software products. Future software products will adhere to the ISQM framework, ensuring compliance before clinical release. (4) Conclusions: This study demonstrates the successful development and implementation of the ISQM program—a standardized framework for managing the quality of in-house software products. The ISQM program ensures continuous and consistent oversight, playing a critical role in guaranteeing safe and high-quality patient care in the field of radiation oncology.

## 1. Introduction

Artificial intelligence (AI) techniques are increasingly being developed and implemented to enhance the safety, quality, and efficiency of workflows in radiation oncology [1,2]. While commercial AI solutions are widely available [3,4], the development and adoption of in-house software are expanding alongside technological advancements [3,5]. Some vendors now offer application programming interfaces (APIs) that allow users to develop and integrate in-house software with commercial systems [5,6,7,8]. In addition to AI-driven applications, radiation oncology also benefits from a wide range of other in-house software products.

Our institution has developed numerous in-house software products utilizing various resources to support clinical tasks such as treatment planning, secondary dose calculation, treatment plan evaluation, chart checks, and more. Traditionally, each software product underwent commissioning before clinical implementation, and quality assurance (QA) programs were developed and executed on an ad hoc basis as part of routine maintenance. As the scope of clinical software development expanded, we established an in-house software quality management (ISQM) program to ensure a standardized QA process and comprehensive documentation. This program enhances the safety, efficiency, and long-term maintainability of all software products used in our clinic. This report provides an overview of the ISQM program and its implementation at our institution.

## 2. Materials and Methods

Typically, software maintenance refers to activities that react to modifications in a software product [9]. However, in radiation oncology, software maintenance extends beyond this reactive approach to include proactive QA that ensures accurate and safe patient care [2,10]. Our ISQM program integrates clinical activities, QA, and documentation into a standard process. The ISQM workflow is illustrated in Figure 1.

### 2.1. Quality Assurance (QA)

Once a software product has been developed and validated, the institution may decide to proceed with its clinical implementation. This step is referred to as the “initiation of clinical implementation” in Figure 1. A newly developed software product often undergoes multiple iterations before reaching a stable version. This process should primarily be addressed during the development and validation phases before the clinical implementation is initiated. Upon approval, the QA process begins. QA consists of planned and systematic activities aimed to ensure that the software product meets established quality requirements [11]. A well-designed QA program should verify the functionality, accuracy, and reproducibility of the software’s performance.

#### 2.1.1. Code Review

Code review is the first step in the QA process within the ISQM framework. This QA activity involves an independent reviewer, not involved in the product’s development, evaluating the software algorithm and source code. The reviewer must have technical expertise to understand the code and identify potential issues.

The code reviewer examines the source code for defects and evaluates its overall organization and documentation, including formats, naming conventions, and comments. It is essential for the reviewer to ensure that the code is well-documented so that anyone with knowledge of the programming language and subject matter can quickly understand the entire codebase. This code review phase ensures the project’s long-term viability in the event of team turnover and supports future updates. Additionally, the code reviewer should have the capability to continue development if the original developer is no longer available within the institution.

Additionally, the reviewer should identify potential edge cases in which the software may not function as intended but which were not addressed by the developer. If deficiencies are found, the product is returned to the developer for correction, followed by another review before advancing to the next phase. If a suspected issue cannot be confirmed during the review, it should be documented and evaluated in subsequent QA stages, such as acceptance testing or commissioning, for verification.

#### 2.1.2. Acceptance Test

Acceptance testing determines whether a software product is suitable for implementation in its current version. The primary goal of acceptance testing is to verify that the product functions without errors and performs according to its specifications and expected behavior [12]. Acceptance testing in the clinical environment serves as a preliminary check before the more rigorous commissioning stage. Institutions may choose to integrate acceptance testing into the commissioning process to streamline a seamless transition to clinical implementation. If the software operates correctly and produces the expected outputs in the clinical environment—where clinical staff interact with it during their workflow—the acceptance test is deemed successful, and the process can proceed to the commissioning phase. However, if functionality issues or errors are identified, the product must return to the development phase for corrections.

#### 2.1.3. Commissioning

Commissioning is the verifying process that ensures that a software product meets clinical requirements for safe and effective clinical operation. This QA step ensures accuracy and safety before the product is deployed for patient care. The commissioning process should include comprehensive tests that thoroughly evaluate typical clinical scenarios, as well as anticipated edge cases. For example, if the software is designed for photon dose calculation, commissioning should involve testing the dose calculation using all variables that could impact the output. In order to confirm the reproducibility, critical commissioning tests should be repeated with the same input. This ensures that the product performs reliably under expected clinical conditions.

The limits of these variables should also be tested for feasibility. For software with user inputs, all input ranges must be validated to ensure that only expected values are accepted and that inputs adhere to the correct format. Commissioning also includes end-to-end testing to evaluate the entire clinical workflow in which the product will be used. This ensures seamless integration into clinical processes and performance as intended in real-world scenarios. The test cases used during commissioning should be documented and retained for reference. This documentation serves as a resource for future re-commissioning, particularly when the product undergoes version updates.

#### 2.1.4. Routine QA

Routine QA is a streamlined version of commissioning, designed to ensure the consistent and accurate performance of a software product. After commissioning is completed, the software product may be released for clinical use, making routine QA unnecessary for the initial version of the release. However, developing a routine QA protocol during the commissioning phase is beneficial, ensuring a structured approach for future QA.

Routine QA should be performed whenever the software is updated to fix issues, introduce new features, or enhance existing functionality. Developers and QA personnel should collaborate to review the changes and determine whether the existing routine QA procedures remain adequate or require modification. QA protocols may evolve alongside software updates. Additionally, routine QA should be conducted when any related systems or environments that impact the software’s functionality are modified. Such changes may include updates to the platform, network configuration, associated data, or clinical procedures. Even if the software itself remains unchanged, these modifications can introduce compatibility or performance issues, necessitating routine QA.

To identify differences between pre- and post-change conditions, it is desirable to use the same set of tests for routine QA. To confirm the reproducibility, tests should be repeated. Routine QA performed in response to an external change in environment or clinical practice should be performed immediately prior to and directly following that change. A comprehensive routine QA program should define the testing frequency, responsible QA personnel, thresholds for each test item, corrective actions, and documentation requirements. Test cases used for routine QA should be retained as a reference for future evaluation and ongoing QA activities.

### 2.2. Documentation

After completing all QA processes, the institution must train users and finalize all necessary documentation. These documents should be stored in a designated shared location that is easily accessible to all users. Once user training and documentation are in place, the software can be released for clinical use. This section outlines the required documentation for the ISQM program, along with detailed descriptions of each document’s purpose and content.

#### 2.2.1. Product Report

The product report serves as the primary document describing the software product. Maintained by ISQM members, it is intended to help clinical staff understand the software’s functionality, performance, and usage. Updates to the product report are necessary if major changes are made to the product. The product report includes the following key sections:Scope and aims: details the clinical objectives addressed by the software, key methods implemented, workflow description, and the potential user base.Technology specifications: outlines the product’s design or data flow, any intellectual property considerations, and potential risk factors related to design, development, implementation, and future modifications.Requirements: identifies the key personnel, including developers, code reviewers, and QA team members, and specifies the necessary resources such as workstations, servers, data storage, hardware, and software.Documents: lists all relevant documents for the product.

#### 2.2.2. Product Manual

The product manual is a procedural guide for users to support their interaction with the software product. Users should review this document before or during training. It should remain accessible for reference during software operation. The manual must provide clear, step-by-step instructions, allowing users to operate the software independently, even without the assistance of an onsite trainer. Updates to the product manual should be considered whenever the software undergoes significant changes or updates to ensure that the instructions remain accurate and relevant.

#### 2.2.3. QA Documents (Procedures and Reports)

QA documents include QA test procedures and their corresponding test results. Each document should specify the date the QA was performed and include the name of the staff who performed QA.

Code Review QA Documentation: Includes a code review checklist with a brief description or result (e.g., pass or fail) for each checklist item. For a new version of the software, a new code review document can be created. However, if the new version involves only minor code changes, an update note can be appended to the original code review report.Commissioning QA Documentation: For the first version of the software product, this document may combine acceptance testing, commissioning, and routine QA into a single report. Each test item should include a detailed procedure or method explaining what to test and how to conduct the test, along with the resulting data or a descriptive outcome.Routine QA Documentation: For new versions of the software, the routine QA report should include details of any new QA procedures performed to evaluate the added features or changes.

This approach ensures comprehensive and well-organized documentation for both initial releases and subsequent updates, supporting consistent quality management over the product’s lifecycle.

#### 2.2.4. Release Note

The release note is a document that outlines new features and bug fixes introduced in each new version of the software. For the initial release, a release note is not necessary, as the product report and product manual already contain all required information. For subsequent versions, a single release note document can be maintained and updated with new entries for each release. It is essential to clearly indicate the version number and release date, along with a summary of the new features and changes, to ensure clear identification of each version.

When a new version of the software is released, the release note and the results of routine QA should be communicated to users. This ensures that users are informed of updates and any potential changes to the software’s functionality or performance.

#### 2.2.5. Version Control

The version control activity involves managing the file names and storage locations of both previous and current versions. This process should be coordinated with the information technology staff to ensure proper organization and accessibility. The version control document should include the version number, software product file name, rationale for the new version, file location, date of release, and developer’s name. Rather than creating a separate document for each new version, it is advisable to maintain a single version control document that is updated with each new release. This consolidated document should retain records of all previous versions, providing a comprehensive history of the software product’s development and updates.

#### 2.2.6. Log

The log document serves as a centralized record of all activities related to the software product. It includes a brief description of each activity, the date it occurred, and the initials of the staff member involved. While some entries in the log may overlap with other documentation, the log provides a concise summary that allows anyone to quickly review and understand the history of the software product’s management. The maintenance of the log relies on the active participation of developers, QA personnel, and document managers. Assigned staff members should record a one-line entry in the log whenever an activity occurs, such as updates, QA events, or issue (or error) resolutions. In the event of an issue (or error), the log document serves as a valuable resource for tracing the history of activities, aiding in troubleshooting, ensuring transparency, and supporting future improvement of the product.

## 3. Results

The ISQM program was successfully developed and retrospectively applied to one of our in-house software products as a pilot implementation. Upon completing all components of the ISQM process, the required documents were created and reviewed by ISQM team members to ensure compliance and completeness

This Results section describes the ISQM process for an in-house software product named “AutoSetupFields” version 1.0 as an example. Previously, planners manually created six setup fields—anterior, posterior, right lateral, left lateral kV, CBCT, and MV home setup fields—to enable imaging at the treatment machine. This process was automated by using Eclipse (Varian Medical Systems, Palo Alto, CA, USA) scripting API (ESAPI) through in-house software developed in C# by a physicist. To ensure quality and reliability, a second physicist with expertise in C# conducted a code review, while another physicist familiar with the planning process performed QA. The code reviewer identified a couple of minor issues, such as the omission of the CBCT field and a typo in the G90/G270 field names. These were promptly corrected by the developer and verified again by the code reviewer. Given the simplicity of this ESAPI tool, the QA physicist combined acceptance testing and commissioning into a single process. Testing involved verifying the script’s functionality across three different treatment machines and all possible patient orientations (e.g., head-first supine, feet-first supine, etc.). A test patient was generated and saved for future reference. The same set of tests was repeated to confirm reproducibility. The routine QA process was designed to repeat these tests in the future. During testing, an issue was discovered: the software failed to execute if an existing setup field was present in the plan. Instead of modifying the script, the ISQM committee decided to document this limitation in the product manual for user awareness and address it in a future release. The product manual provided step-by-step instructions, including how to launch the software, the directory to access it, and the necessary user actions. The log file recorded three key events: (1) [Date]—V1.0 released for commissioning/QA, (2) [Date]—identification of the software error related to pre-existing setup fields, with a decision to document it in the manual, and (3) [Date]—V1.0 released for clinical use. All documentation was prepared collaboratively by the developer, code reviewer, and QA physicist. So far, the software has not required an updated release.

To ensure consistent documentation, a standardized template for product reports was developed in a question-and-answer format. Developers are required to complete this template by filling in the designated fields with relevant information. This structured format enables readers to quickly locate and understand essential details about each product. A standardized code review checklist was also created to guide code reviewers during comprehensive code review. The checklist includes, at a minimum, all the elements detailed in the Methods section. Figure 2 shows an example product report using the template for the example ESAPI software described in the previous section. Table 1 shows the standard code review checklist, along with example responses for the same script featured in Figure 2.

After preparing a full set of documents for the pilot product, the ISQM program was applied to all other existing in-house software products, ensuring that each met all program requirements. For upcoming software products currently under development or approaching clinical implementation, developers and QA personnel have been informed of the mandatory adherence to the ISQM program. Compliance with all ISQM requirements is now a prerequisite for releasing any new software products for clinical use.

## 4. Discussion

The ISQM program was initially developed to manage Eclipse script automation tools. However, its scope has since expanded to encompass a wide variety of applications. The most immediate candidates for inclusion are scripts utilizing the API of the treatment planning system. Beyond this, the ISQM program has been adapted to include any custom code implemented by members of our department that could influence clinical decision making. These applications range from simple tools, such as expressions within a spreadsheet or custom database queries, to more complex systems, like AI-based treatment planning tools. By broadening the program’s scope, we ensure that all in-house software products, regardless of complexity, adhere to a consistent quality management framework. This comprehensive approach enhances safety and reliability, while also supporting the seamless integration of diverse software products into clinical workflows.

Once all QA tests are completed and all required documents are prepared, the in-house software product is released for clinical use, as indicated by “routine clinical use” in Figure 1. From this point forward, routine QA is performed when triggered by specific events such as the following:Platform or network configuration changes;Modifications to associated data or clinical practice;Discovery of bugs;User feedback requesting upgrades;Other significant updates.

Effective communication within the clinical team—including developers, QA personnel, IT staff, and end users—is crucial to promptly identify and address triggering events. If a routine QA test fails or raises concerns, developers and managing staff must evaluate and update the product as necessary. Otherwise, the product may continue to be used in clinical workflows. When a software product is updated, it re-enters the development and validation phases, following the ISQM workflow beginning with code review. Depending on the extent of the changes in a new version, certain steps—such as acceptance testing and commissioning—may be omitted, provided that routine QA sufficiently verifies the updates. Routine QA itself can be modified to efficiently and effectively address the specific changes in the updated version. Additionally, all relevant documents—including the QA document, release note, version control document, and log—should be updated to reflect the new version and associated changes.

The integration of AI techniques in radiation oncology offers significant potential to improve efficiency, reliability, and scalability. AI-driven tools can automate various aspects of the clinical workflow, enhancing overall performance. The design and development of AI tools vary based on their intended applications. This report specifically focuses on the maintenance and quality management of in-house AI tools rather than on their design or implementation. A thorough understanding of each AI tool or software is essential for developing effective QA processes. However, discussions on different AI applications or software designs are beyond the scope of this report.

The ISQM program was developed to address the unique challenges of managing in-house software products in radiation oncology. While Moran et al. [13] provide a broad framework for implementing and managing healthcare software, our manuscript focuses specifically on a detailed, step-by-step framework tailored for in-house software products in clinical radiation therapy. This targeted approach emphasizes the consistent application of quality assurance, documentation, and maintenance processes to ensure safety, accuracy, and long-term viability—essential in this high-stakes field. The ISQM program introduces a comprehensive integration of code review, acceptance testing, commissioning, and routine QA into a unified framework. This level of technical specificity is vital for aligning in-house software with clinical requirements and facilitating seamless integration into patient care workflows. The ISQM program requires adequate expertise within the institution to fully staff the code reviewer and developer roles. Loss of this expertise or personnel could result in disruptions in patient care if critical software cannot be maintained. Our study addresses key challenges associated with custom software in clinical settings, such as maintaining compatibility with evolving platforms, managing developer turnover, and ensuring comprehensive documentation to support continuity and user training. By providing a structured and scalable solution, the ISQM program supports the safe, effective, and reliable use of in-house software in clinical practice.

Most studies on AI software development and implementation primarily focus on feasibility, validation, or commissioning, including QA [5,14,15,16]. However, this ISQM program extends beyond these aspects to emphasize the transparency, continuity, and longevity of software products. This report outlines the work processes and requirements necessary for the consistent management of in-house software products. We anticipate several challenging scenarios that could disrupt clinical workflows involving in-house software products. Examples include software becoming incompatible with updated clinical environments (e.g., changes in computer operating systems), the departure of the original developer from the institution, or the need to train new staff members. The ISQM program is specifically designed to address these challenges, ensuring the sustained and consistent management of all in-house developed software products over time.

## 5. Conclusions

In-house software products offer significant benefits to clinics by delivering customized solutions that enhance the efficiency, consistency, and quality of performance. However, their clinical implementation and maintenance require collaborative efforts and a strong commitment to quality and safety. This study demonstrates the successful development and implementation of the ISQM program, a standardized framework designed for the quality management of in-house software products. The ISQM program ensures continuous and consistent oversight of all software products. It plays a critical role in supporting safe and high-quality patient care in the field of radiation oncology.

## Figures and Tables

**Figure 1 bioengineering-12-00352-f001:**
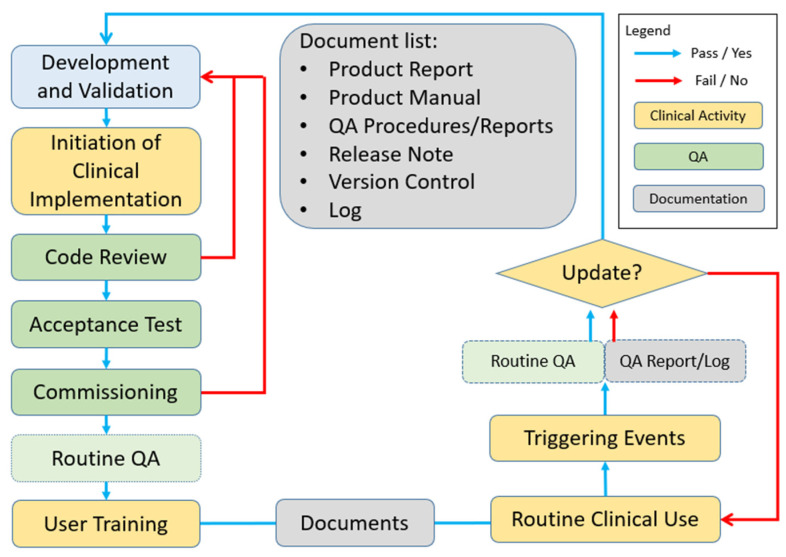
ISQM process flowchart.

**Figure 2 bioengineering-12-00352-f002:**
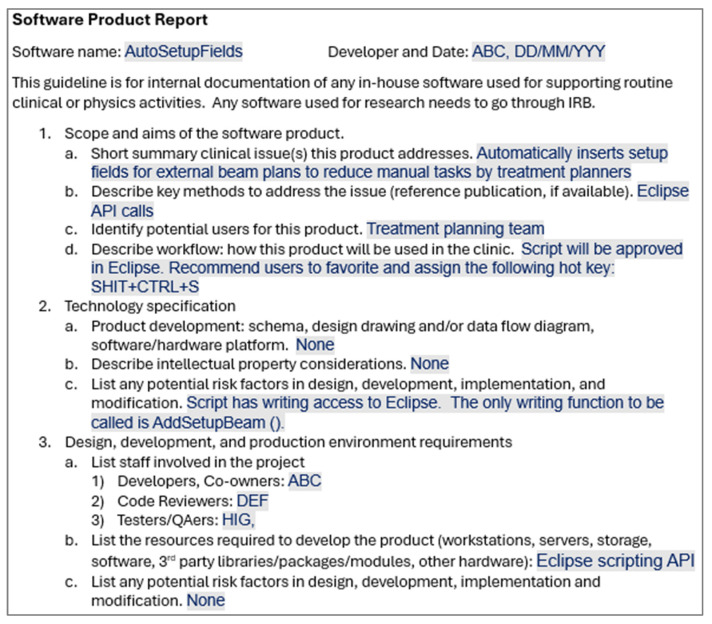
Product report template filled out using an in-house software product “AutoSetupFields” as an example. Staff names and the date of the report are blinded.

**Table 1 bioengineering-12-00352-t001:** Code review checklist. Each item to be answered Yes/No (or Pass/Fail). Notes to be made, if needed.

Software name: AutoSetupFields Code reviewer name: DEF Date: DD/MM/YYYY	
1	Administrative (pass required to proceed)
1-a	Is the code in ISQM directory? Yes
1-b	Does the code run? Yes (note: ran is as ‘Approval for Test’)
2	Business Requirements Documentation (Pass required to proceed)
2-a	Is the code pulling from the appropriate tables (live or data warehouse)? Yes
2-b	Is the code logic sound? Yes
3	Performance tuning/optimization
3-a	Does the code pull only relevant data? (e.g., no extraneous data?) Yes
3-b	Is the run time acceptable? (e.g., if the output is real-time data, is the runtime acceptable?) Yes
4	Naming convention
4-a	Are the variable or class names readily understood? Yes
4-b	Are the function or method names readily understood? Yes
4-c	Is the code readable to a non-expert? Yes
5	Formatting
5-a	Was the code run through a commonly agreed upon formatter with commonly agreed upon parameters? Yes
6	Comments and Header
6-a	Does the code include a header to document information about the code: script name/s, purpose, date, author and change log on significant updates of the code logic? NA
6-b	Does the code include comments to explain the logic, where appropriate, on the different sections of the script? Yes
7	Code Quality
7-a	Identify edge case for QA Yes (note: no CBCT field; Typo in the G90/G270 fields)
7-b	Are user-input ranges checked? Can invalid input be handled properly? NA
7-c	Are user-input files validated? NA
7-d	External packages (or 3rd party library of code) are used? NA

## Data Availability

Data are available upon request.

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
