# Peer review of "Quality Management for Radiation Oncology In-House Software Products"

_bioengineering, 2025, doi:10.3390/bioengineering12040352_

Round 1
Reviewer 1 Report
Comments and Suggestions for Authors
The authors described their recommendation how to check and validate an in-house software
The study is very simple but well define and they clarified all steps.
In this AI era, I found it interesting and they suggested a good schema that I can also recommend to follow.
I agree about they wrote and I don't have any comment. For me it is ready to be published
Comments on the Quality of English Languagefor me it is ok
Author Response
Thank you for the review and positive feedback.
Reviewer 2 Report
Comments and Suggestions for Authors
This manuscript has presented the study of ISQM, which is called In-house Software Quality Management program. I agree with the authors’ statements and general rules of ISQM and understand the study will play critical roles in the radiation Oncology. It is pity if the author can conclude or manuscript is written combined with the practical ISQM software development which is now already used in the clinical deployment process that would be very nice. From the statements and description, we can understand the standardization of ISQM and its importance, but it appears a bit empty not so practical.
- The authors have mentioned the AI techniques and its applications, we all know its importance ,especially in the application of radiation oncology, the authors haven’t expanded or explored the possibility, viability, feasibility and so on, I hope the authors could emphasize it in the application of ISQM program design.
- I agree with the authors’ quality assurance design and control process. About the reproducibility of the software product’s performance, I don’t know how to validate this problem?
- Line 69, the code reviewer should also be capable of continuing its development if the original developer is no longer available within the institution. I am not completely sure this could be done in the real project. Maybe the best thing we could do is to keep the software developer stably work in the group.
- Line 85-86, However, if functionality issues or errors are identified during acceptance testing, the product must return to the development phase for corrections. This would be happened very often for the new software developing process. I think it is better to keep the software running in the commissioning stage for a while until without any problems. As the authors’ statement: “This QA step ensures accuracy and safety for clinical use before the product is deployed for actual patient care.”;’ After commissioning is completed the software product may be released for clinical use, making routine QA unnecessary for the initial version of the release.’
Author Response
Comments1: The authors have mentioned the AI techniques and its applications, we all know its importance ,especially in the application of radiation oncology, the authors haven’t expanded or explored the possibility, viability, feasibility and so on, I hope the authors could emphasize it in the application of ISQM program design.
Response1: Thank you for the critical feedback. We agree that this manuscript does not explore viability or feasibility of AI techniques in relation to ISQM program. We added a paragraph in the discussion to address this point. Even if we are not addressing design aspect of AI techniques (possibility, viability and feasibility) in this report, it is critical for QA personnel to understand AI application thoroughly for successful ISQM program. Please see the 3d paragraph of the discussion. (See lines 277 – 284 in the clean version )
Comments2: I agree with the authors’ quality assurance design and control process. About the reproducibility of the software product’s performance, I don’t know how to validate this problem?
Response2: Yes, we agree. Ideally, we hope our software developers to stay, but they might leave to other institutions and we need to make sure we have staff who can update existing software. Code reviewers have that potential. We modified the discussion to address this issue. ISQM program requires adequate expertise within the institution to fully staff the code reviewer and developer roles. Loss of this expertise or personnel could result in disruptions in patient care if critical software cannot be maintained. Our study addresses key challenges associated with custom software in clinical settings, such as maintaining compatibility with evolving platforms, managing developer turnover, and ensuring comprehensive documentation to support continuity and user training. (see lines 295-301 in the clean version)
Comments3: Line 69, the code reviewer should also be capable of continuing its development if the original developer is no longer available within the institution. I am not completely sure this could be done in the real project. Maybe the best thing we could do is to keep the software developer stably work in the group.
Response3: Yes, we agree. Ideally, we hope our software developers to stay, but they might leave to other institutions and we need to make sure we have staff who can update existing software. Code reviewers have that potential. We modified the discussion to address this issue. ISQM program requires adequate expertise within the institution to fully staff the code reviewer and developer roles. Loss of this expertise or personnel could result in disruptions in patient care if critical software cannot be maintained. Our study addresses key challenges associated with custom software in clinical settings, such as maintaining compatibility with evolving platforms, managing developer turnover, and ensuring comprehensive documentation to support continuity and user training. (see lines 295-301 in the clean version)
Comments4: Line 85-86, However, if functionality issues or errors are identified during acceptance testing, the product must return to the development phase for corrections. This would be happened very often for the new software developing process. I think it is better to keep the software running in the commissioning stage for a while until without any problems. As the authors’ statement: “This QA step ensures accuracy and safety for clinical use before the product is deployed for actual patient care.”;’ After commissioning is completed the software product may be released for clinical use, making routine QA unnecessary for the initial version of the release.’
Response4: Very good point. We updated the report accordingly. A newly developed software product often undergoes multiple iterations before reaching a stable version. This process should primarily be addressed during the development and validation phases before the clinical implementation is initiated. (see Lines 55 – 58 in the clean version) Acceptance testing in the clinical environment serves as a preliminary check before the more rigorous commissioning stage. Institutions may choose to integrate acceptance testing into the commissioning process to streamline a seamless transition to clinical implementation. (see Lines 85-88 in the clean version)